# Unravel Structured Heterogeneity of Tasks in Meta-Reinforcement Learning via Exploratory Clustering

## Abstract

Meta-reinforcement learning (meta-RL) is developed to quickly solve new tasks by leveraging knowledge from prior tasks. The assumption that tasks are drawn IID is typically made in previous studies, which ignore possible structured heterogeneity of tasks. The non-transferable knowledge caused by structured heterogeneity hinders fast adaptation in new tasks. In this paper, we formulate the structured heterogeneity of tasks via clustering such that transferable knowledge can be inferred within different clusters and non-transferable knowledge would be excluded across clusters thereby. To facilitate so, we develop a dedicated exploratory policy to discover task clusters by reducing uncertainty in posterior inference. Within the identified clusters, the exploitation policy is able to solve related tasks by utilizing knowledge shared within the clusters. Experiments on various MuJoCo tasks showed the proposed method can unravel cluster structures effectively in both rewards and state dynamics, proving strong advantages against a set of state-of-the-art baselines.

## 1 Introduction

Conventional reinforcement learning (RL) is notorious for sample inefficiency, which often requires millions of interactions with the environment to learn a performing policy for a new task. Inspired by the human learning process, meta-reinforcement learning (meta-RL) is proposed to quickly learn new tasks by leveraging knowledge shared by related tasks (Finn et al., 2017; Duan et al., 2016; Wang et al., 2016). Extensive efforts have been put into learning and utilizing transferable knowledge in meta-RL. For example, Finn et al. (2017) proposed to learn a set of shared meta parameters which is used to initialize the local policy when a new task comes. Duan et al. (2016) and Wang et al. (2016) trained an RNN encoder to characterize prior tasks according to the interaction history.

However, little attention has been paid to the situations where some knowledge is only locally transferable among tasks. All the aforementioned methods implicitly assume tasks have substantially shared structures, and thus knowledge can be broadly shared across all tasks. However, heterogeneity among tasks exists in practice, if not prevails, which hampers the effectiveness of existing meta-RL algorithms. For example, the necessary skills for the Go game can hardly be applied to the Gomoku game, though both of them operate on the same chessboards. We formulate this scenario as a more complicated but also more realistic meta-RL setting where tasks are originated from different distributions, i.e., tasks are clustered. As a result, some knowledge is locally transferable within clusters, but cannot be shared globally. We refer to this as *structured heterogeneity* among RL tasks, and explicitly model cluster structures in the task distribution to capture cluster-level knowledge [1].

Structured heterogeneity has been studied in supervised meta-learning (Yao et al., 2019); but it is a lot more challenging to be handled in meta-RL, where the key bottleneck is *how to unravel the clustering structure in a population of RL tasks*. This can be further decomposed into two key research questions, namely population-level structure estimation and task-level inference. Different from supervised learning tasks where static task-specific data is already provided before meta learning, the observations in RL tasks are collected by an agent's interactions with the environment. As

---

[1]In this paper, we do not assume the knowledge in different clusters is exclusive, and thus each cluster can also contain overlapping global knowledge, e.g., motor skills in locomotion tasks.

a result, in addition to the original explore-exploit trade-off an RL agent needs to handle for return maximization, now a meta-RL agent also has to balance the trade-off between the need of clustering structure identification and the need of return maximization when taking actions. Oftentimes, the action benefiting return maximization does not necessarily help clustering structure identification. The situation becomes more pressing when solving new tasks at test time, where the agent needs to differentiate what is transferable at the cluster-level, task-level, or not at all, for fast adaptation.

To handle the structured heterogeneity of tasks in meta-RL, we propose a cluster-based meta-RL algorithm, called MILET: **M**eta re**I**nforcement **L**earning via **E**xploratory clus**T**ering, which is designed to explore clustering structures of tasks and enhance fast adaptation on new tasks with cluster-level transferable knowledge. Specifically, we perform cluster-based posterior inference (Rao et al., 2019; Dilokthanakul et al., 2016) to infer the clustering structure of a new task. To accelerate cluster inference, we learn a dedicated exploration policy that is aware of cluster structures and is trained to explore the environment to reduce uncertainty in cluster inference as fast as possible. Furthermore, we design a composed reward function for it to encourage a coarse-to-fine exploratory strategy. New tasks can be quickly adapted with the explored cluster information by using locally transferable knowledge within clusters. We compare our method with a rich set of state-of-the-art meta-RL baselines on various MuJoCo environments (Todorov et al., 2012) with cluster structures in both rewards and state dynamics. We also show our method can mitigate the sparse reward issue by sample-efficient exploration on cluster structures, which provides hints to solve a specific task. We further test our method on environments without explicit clustering structures, the results show our method can automatically discover locally transferable knowledge and benefit the adaptation.

## 2    RELATED WORK

**Task modeling in meta-learning.** Task modeling is important to realize fast adaptation in new tasks in meta learning. Finn et al. (2017) first proposed the model-agnostic meta learning (MAML) aiming to learn a shared model initialization, i.e., the meta model, given a population of tasks. MAML does not explicitly model tasks, but it expects the meta model to be only a few steps of gradient update away from all tasks. Later, an array of methods extend MAML by explicitly modeling tasks using given training data in supervised meta-learning. Lee & Choi (2018) learned a task-specific subspace of each layer's activation, on which gradient-based adaptation is performed. Vuorio et al. (2019) explicitly learned task embeddings given data points from each task, and then used it to generate task-specific meta model. Yao et al. (2019) adopted a hierarchical task clustering structure, which enables cluster-specific meta model. Such a design encourages the solution to capture locally transferable knowledge inside each cluster, similar to our MILET model. However, task information is not explicit in meta-RL: since the true reward/state transition functions are not accessible by the agent, the agent needs to interact with the environment to collect observations about the tasks, while maximizing the return from the interactions. MILET performs posterior inference of a task's cluster assignment based on its ongoing trajectory; better yet, it is designed to behave exploratorily to quickly identify tasks' clustering structures.

**Exploration in meta-reinforcement learning.** Exploration plays an important role in meta-RL, as the agent can only learn from its interactions with the environment. In gradient-based meta-RL (Finn et al., 2017), the local policy is trained on the trajectories collected by the meta policy, thus the exploration for task structure is not explicitly handled. Stadie et al. (2018) and Rothfuss et al. (2019) computed gradients w.r.t. the sampling distribution of the meta policy, in addition to the collected trajectories. Gurumurthy et al. (2020) learned a separate exploration policy for MAML to collect informative pre-update trajectories. Gupta et al. (2018) also extended MAML by using learnable latent variables to control different exploration behaviors. The context-based meta-RL algorithms (Duan et al., 2016; Wang et al., 2016) automatically learn to trade-off exploration and exploitation by learning a policy conditioned on the current context. Zintgraf et al. (2020) explicitly provided the task uncertainty to the policy to facilitate exploration via variational inference. Zhou et al. (2019) introduced intrinsic rewards encouraging exploration that can improve the prediction of dynamics. Zhang et al. (2021) and Liu et al. (2021) developed a separate exploration policy by maximizing the mutual information between task ids and inferred task embeddings. However, all the aforementioned exploration methods are oblivious to the structured heterogeneity of tasks, leading to inefficient exploration and ignorance of locally transferable knowledge within clusters.

MILET learns the exploration policy to explicitly reduce the uncertainty in cluster inference, which is important to utilize locally transferable knowledge.

## 3 BACKGROUND

**Meta-reinforcement learning.** We consider a family of Markov decision processes (MDPs) [2] $p(\mathcal{M})$, where an MDP $M_i \sim p(\mathcal{M})$ is defined by a tuple $M_i = (\mathcal{S}, \mathcal{A}, R_i, T_i, T_{i,0}, \gamma, H)$ with $\mathcal{S}$ denoting its state space, $\mathcal{A}$ as its action space, $R_i(r_{t+1}|s_t, a_t, s_{t+1})$ as its reward function, $T_i(s_{t+1}|s_t, a_t)$ as its state transition function, $T_{i,0}(s_0)$ as its initial state distribution, $\gamma$ as a discount factor, and $H$ as the horizon of an episode. The index $i$ represents the task id, which is provided to agents in some works (Zhang et al., 2021; Liu et al., 2021; Rakelly et al., 2019). Following Zintgraf et al. (2020), we consider a more general setting where the task id is unavailable to the agent, since it becomes meaningless when we keep encountering new tasks. Tasks sampled from $p(\mathcal{M})$ typically differ in the reward and/or transition functions. For an MDP, we run a *trial* consisting of $N + 1$ episodes (Duan et al., 2016). Following the evaluation settings in previous works (Finn et al., 2017; Liu et al., 2021; Zhang et al., 2021; Rothfuss et al., 2019), the first episode in a trial is reserved as an *exploration* episode to gather task-related information, and an agent is evaluated by the returns in the following $N$ *exploitation* episodes.

Inside a trial, we denote the agent's interaction with the MDP at time step $n$ as $\tau_n = \{s_n, a_n, r_n, s_{n+1}\}$, and $\tau_{:t} = \{s_0, a_0, r_0, ..., s_t\}$ denotes the whole interaction history collected before time $t$. In the exploration episode, an agent should form the most informative history $\tau^\psi$ by rolling out an exploration policy $\pi_\psi$ parameterized by $\psi$. In exploitation episodes, the agent executes the exploitation policy $\pi_\phi$ parameterized by $\phi$ (in some prior work, $\pi_\psi$ and $\pi_\phi$ could be the same) conditioned on $\tau^\psi$ and, optionally, the history collected in exploitation episodes $\tau^\phi$. The agent's goal is to maximize the returns in exploitation episodes, which can be formally expressed as,

$$\mathcal{J}(\pi_\psi, \pi_\phi) = \mathbb{E}_{M_i \sim p(\mathcal{M}), \tau^\psi \sim \pi_\psi} \Big[ \sum_{t=0}^{N \times H} R_i\big(\pi_\phi(\tau^\psi; \tau_{:t}^\phi)\big) \Big], \tag{1}$$

where $R_i\big(\pi_\phi(\tau^\psi; \tau_{:t}^\phi)\big)$ is the return of $\pi_\phi$ conditioned on $\tau^\psi$ and $\tau_{:t}^\phi$ at time step $t$ in task $M_i$.

**Mixture of RL tasks.** In this paper, we consider a more genearl and realistic setting, where different task distributions form a mixture,

$$p(\mathcal{M}) = \sum_{c=1}^{C} w_c \cdot p_c(\mathcal{M}), \tag{2}$$

where $C$ is the number of mixing components (i.e., clusters) and $w_c$ is the corresponding weight of component $c$, such that $\sum_{c=1}^{C} w_c = 1$. Every task is sampled as follows,

- Sample a cluster $c$ according to the multinomial distribution of $Mul(w_1, ..., w_C)$;
- Sample a reward function $R$ or a transition function $T$ or both from $p_c(\mathcal{M})$.

The knowledge shared in different clusters could be different. For example, two clusters of target positions can exist in a navigational environment and each task is associated with a particular target position. In the first cluster, the target positions concentrate in the upper part of the map, while the target positions in the second cluster fall in the bottom part. The knowledge that how the agent moves to the upper part in the first cluster should not be transferred to the second cluster; but it is crucial for different tasks in the first cluster. Such locally transferable knowledge cannot be effectively captured by previous meta-RL solutions, since they are only designed to realize globally transferable knowledge. Our method is able to discover and utilize such local knowledge by modeling the underline structures of tasks distributions.

## 4 METHODOLOGY

In this section, we present MILET in detail. Firstly, we introduce how to estimate population-level task structures using the collected trajectories via cluster-based variational inference. Then,

---

[2]The terms of environment, task and MDP are used interchangeably in this paper.

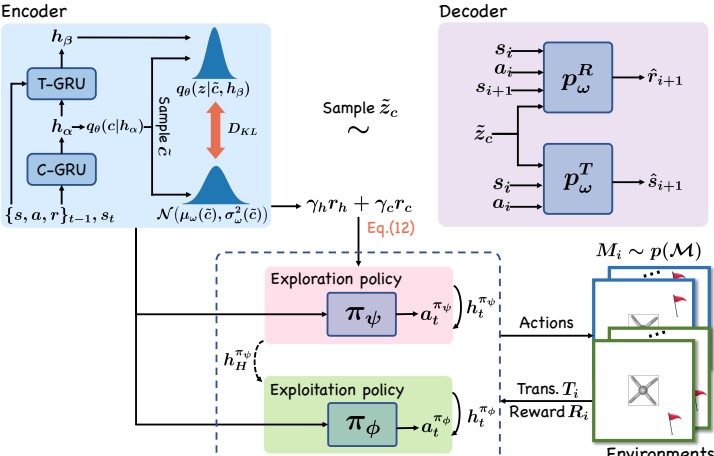

Figure 1: MILET architecture. The encoder processes ongoing trajectories and performs cluster-based task inference $q_\theta(z|c, h_\beta)$. The exploration policy $\pi_\psi$ is trained to explore the environment to find the most certain cluster assignment $c$. The explored information is passed to the exploitation policy $\pi_\phi$ to facilitate fast adaptation in the inferred task $M_{z_c}$.

we explain the exploration policy trained by the exploration-driven reward, which is designed to quickly identify the cluster of a new task. In particular, in each task we first execute the exploratory policy to collect the coarse-grained cluster information; then we adapt the task policy with the help of the inferred posterior cluster distribution to obtain fine-grained task information. The architecture of MILET is shown in Figure 1.

## 4.1 CLUSTER-BASED VARIATIONAL INFERENCE WITH CONSISTENCY REGULARIZATION

Since the reward and transition functions are unknown to the agent, we learn a stochastic latent variable $c_i$ to infer the cluster assignment of current task $M_i \sim p_c(\mathcal{M})$. Based on $c_i$, we infer another stochastic latent variable $z_i$ carrying task-level information, i.e., reward/transition functions that define the task. For simplicity, we first drop the script $i$ in this section as we will only use one task as an example to illustrate our model design. We believe all information about the current task is encoded in $\{z, c\}$, e.g., once we know the cluster and task assignments, we can perfectly reconstruct the dynamics of the task, i.e., the trajectories. When a new task arrives, we decode its characteristics by the posterior distribution $p(z, c|\tau_{:t})$ with respect to the interaction history up to time $t$. The inferred task information $z_c$ ($z$ conditioned on $c$) is then input into the policy $\pi_{\psi/\phi}(a_t|s_t, z_c)$.

However, the exact posterior $p(z, c|\tau_{:t})$ defined by Eq.(2) is intractable. Instead, we learn an approximated variational posterior $q_\theta(z, c|\tau_{:t}) = q_\theta(z|\tau_{:t}, c)q_\theta(c|\tau_{:t})$, in which we learn two dependent inference networks and collectively denote their parameters as $\theta$. On top of the inference networks, we learn a decoder $p_\omega$ to reconstruct the collected trajectories. The whole framework is trained by maximizing the following quantity,

$$\mathbb{E}_{\rho(\mathcal{M}, \tau^+)} \Big[ \log p(\tau^+|\pi) \Big], \tag{3}$$

where $\rho$ is the distribution of trajectories induced by the policies $\pi = \{\pi_\psi, \pi_\phi\}$ and the environment, and $\tau^+ = \{\tau^\psi, \tau^\phi\}$ denotes all the trajectories collected in a trial, the length of which is denoted as $H^+ = (N+1)H$. Note that we use trajectories from both exploration and exploitation episodes since the rewards and state transitions in the same task are generated by the same underlying MDP. We omit the dependencies on $\pi$ to simplify our notations in later discussions. Instead of optimizing the intractable objective in Eq.(3), we optimize its evidence lower bound (ELBO) w.r.t. the approximated posterior $q_\theta(z, c|\tau_{:t})$ estimated via Monte Carlo sampling (Rao et al., 2019) (full derivation can be found in Appendix A),

$$ELBO_t = \mathbb{E}_\rho \Big[ \overbrace{\mathbb{E}_{q_\theta(z, c|\tau_{:t})} \big[ \ln p_\omega(\tau^+|\tilde{z}_c) \big]}^{\text{cluster-specific reconstruction loss}} - \overbrace{\mathbb{E}_{q_\theta(c|\tau_{:t})} \big[ \text{KL}(q_\theta(z|c, \tau_{:t}) \| p_\omega(z|c)) \big]}^{\text{cluster-specific regularization}}$$

$$- \underbrace{\text{KL}(q_\theta(c|\tau_{:t}) \| p(c))}_{\text{categorical regularization}} \Big], \tag{4}$$

where $p_\omega(z|c) = \mathcal{N}\big(\mu_\omega(c), \sigma_\omega^2(c)\big)$ is a learnable cluster-specific prior, which is different from the simple Gaussian prior used in single-component VAE (Kingma & Welling, 2013). This prior allows the model to capture the unique structure of each cluster. Its parameters are included in $\omega$ since the cluster structure is also a part of the environment. $\tilde{z}_c$ is the latent variable sampled from $q_\theta(z|c, \tau_{:t}) = \mathcal{N}\big(\mu_\theta(c, \tau_{:t}), \sigma_\theta^2(c, \tau_{:t})\big)$, using the reparameterization trick (Kingma & Welling, 2013). $q_\theta(c|\tau_{:t})$ outputs the approximated posterior cluster distribution given $\tau_{:t}$ [3]. $p(c)$ is a fixed non-informative multinomial distribution representing the prior cluster distribution of tasks.

Similar to Zintgraf et al. (2020), the first term $\ln p_\omega(\tau^+|\tilde{z}_c)$ in Eq.(4) can be further factorized as

$$\ln p_\omega(\tau^+|\tilde{z}_c, \pi) = \ln p(s_0|\tilde{z}_c) + \sum_{i=0}^{H^+-1} \bigg[ \ln p_\omega(s_{i+1}|s_i, a_i, \tilde{z}_c) + \ln p_\omega(r_{i+1}|s_i, a_i, s_{i+1}, \tilde{z}_c) \bigg]$$

$$\approx const. + \sum_{i=0}^{H^+-1} -(r_i - \hat{r}(s_i, a_i, s_{i+1}, \tilde{z}_c))^2 - \lambda_s \|s_{i+1} - \hat{s}(s_i, a_i, \tilde{z}_c)\|_2^2, \tag{5}$$

where $p(s_0|\tilde{z}_c)$ is the initial state distribution in a task, and we consider it as a constant by assuming identical distribution of the initial states across clusters. The second and third terms are likelihood derived from decoders for transition and reward functions. $\lambda_s$ control the effect of the state transition in variational inference. The density function of $p_\omega(s_{i+1}|s_i, a_i, \tilde{z}_c)$ and $p_\omega(r_{i+1}|s_i, a_i, s_{i+1}, \tilde{z}_c)$ is difficult to estimate in continuous state and action spaces. Denote the corresponding decoder outputs as $\hat{r}$ and $\hat{s}$, we use L2 distance to approximate the log-likelihood functions (Zhang et al., 2021).

In the inference networks $q_\theta(z|\tau_{:t}, c)$ and $q_\theta(c|\tau_{:t})$, we follow Duan et al. (2016); Zintgraf et al. (2020) to encode the history $\tau_{:t}$ by Gated Recurrent Units (GRUs) (Chung et al., 2014). We propose a stacked GRU structure (shown in Figure 1) to differentiate the information for cluster and task inference in the hidden space. Specifically, we set a task GRU (T-GRU) and a cluster GRU (C-GRU), both of which encode the history $\tau_{:t}$, but with different levels of granularity. T-GRU is set to capture fine-grained task-specific patterns in the history as it is optimized to reconstruct trajectories of a specific task. C-GRU captures more coarse-grained patterns as it is set to help T-GRU reconstruct all trajectories within a cluster. To realize this difference, the output $h_\beta$ of T-GRU is only provided to $q_\theta\big(z|h_\beta(\tau_{:t}, h_\alpha), c\big)$, while the output $h_\alpha$ of C-GRU is passed to both cluster inference $q_\theta\big(c|h_\alpha(\tau_{:t})\big)$ and task inference $q_\theta\big(z|h_\beta(\tau_{:t}, h_\alpha), c\big)$. This also reflects our dependency assumption about the task structure. We denote $h = \{h_\alpha, h_\beta\}$, and $h$ is passed between episodes in a trial.

Different from the static training data provided beforehand in supervised meta-learning settings, the trajectory data is incrementally collected by the agent in meta-RL, which brings both challenges and opportunities in obtaining the most informative information for structure identification. Firstly, inside a trial, the structure inference improves as more interactions are collected, which means the agent's belief about the task structure could change thereby. This introduces a contradiction since the inference result should stay consistent within a given task, no matter how trajectory changes over episodes. We attribute this property as *in-trial* consistency. It can be measured by $\mathrm{KL}(q(c|\tau_{:t_1}) \| q(c|\tau_{:t_2}))$, where $t_1$ and $t_2$ refer two arbitrary timesteps in a trial and the KL divergence measures the inconsistency between two cluster posterior distributions. We realize the notion of cluster inference consistency via the following regularizer,

$$\mathcal{L}_\mathrm{I} = \frac{1}{H^+-1} \sum_{t=0}^{H^+-1} \mathrm{KL}\big(q_\theta(c|\tau_{:t}) \| q_\theta(c|\tau_{:t+1})\big). \tag{6}$$

This regularizer minimizes the inconsistency between any two consecutive time steps inside a trial. The consistency of task inference is implicitly enforced once the cluster inference becomes consistent. Furthermore, $q_\theta(c|\tau_{:t})$ is encouraged to find structures beyond the task-level by this regularizer, i.e., cluster structures, as in-trial cluster inference is enforced to be the same.

Similarly, since the cluster-specific prior $p_\omega(z|c)$ is learnable, the task inference can become inconsistent if the prior changes drastically across training epochs. More seriously, oscillation in the inference of latent variable $z$ can cause the collapse of policy training, as tasks across clusters might

---

[3] In practice, we use the Gumbel-softmax trick to simplify the calculation.

be assigned with the same latent variable $z$. We conclude it as the *prior* consistency property and realize it via the following regularization,

$$\mathcal{L}_{\mathrm{P}} = \frac{1}{C} \sum^{C} \mathrm{KL}\big(p_\omega(z|c) \,\|\, p_{\mathrm{tgt}}(z|c)\big), \tag{7}$$

where $p_{\mathrm{tgt}}(z|c)$ is a target network and its parameters are the same as $p_\omega(z|c)$ but updated in a much slower pace. We finally conclude the objective in our cluster-based variational inference (CBVI),

$$\mathcal{L}(\theta, \omega) = \mathbb{E}_{p(\mathcal{M})}\Big[ \sum_{t=0}^{H+} ELBO_t - \lambda_{\mathrm{I}}\mathcal{L}_{\mathrm{I}} - \lambda_{\mathrm{P}}\mathcal{L}_{\mathrm{P}} \Big], \tag{8}$$

where $\lambda_{\mathrm{I}}$ and $\lambda_{\mathrm{P}}$ are hyper-parameters to control the strength of two regularizers.

### 4.2 Exploration for Reducing Task Inference Uncertainty

The adaptation procedure in a new task can be decomposed into two sub-goals: (1) explore clustering structure; (2) solving the task with the inferred task information. These two sub-goals are not in parallel, but the correct clustering structure identification is the prerequisite for maximizing task return. Thus, we are motivated to learn two separate policies as shown in Figure 1. One takes exploratory behaviors to collect cluster and task information, i.e., the exploration policy $\pi_\psi$. The other is optimized to solve the task with the collected information, i.e., the exploitation policy $\pi_\phi$.

Clustering structures provide hints to solve specific tasks. We train a separate exploration policy to provide a good basis for task-solving by exploring the environment to quickly identify the cluster assignment of a task. We evaluate the quality of exploration using two principles. Firstly, *whether the trajectory of an exploration episode can reduce the uncertainty of the cluster inference.* Secondly, *whether the inference result is consistent.* The exploration trajectory should lead to a consistent cluster, other than switching among clusters. We conclude them as *certain* and *consistent* exploration.

We design two intrinsic rewards to encourage certain and consistent behaviors. First, we use the entropy of cluster inference network $q_\theta(c|\tau_{:H}^\psi)$ to measure the *uncertainty* of the inferred cluster. For a new task, we look for trajectories that provide the most certain cluster inference. We formalize the objective is as follows, omitting the subscript $\theta$ and superscript $\psi$ for simplicity,

$$H(q(c|\tau_{:H})) = -\mathbb{E}\big[\ln q(c|\tau_{:H})\big] = -\mathbb{E}\Big[\ln q(c|\tau_0) + \sum_{t=0}^{H-1} \ln \frac{q(c|\tau_{:t+1})}{q(c|\tau_{:t})}\Big]. \tag{9}$$

We then define an intrinsic reward of each action by telescoping the second term similar to Zhang et al. (2021); Liu et al. (2021),

$$r_h(a_t) = \mathbb{E}\Big[\ln \frac{q(c|\tau_{:t+1} = [s_{t+1}; a_t; r_t; \tau_{:t}])}{q(c|\tau_{:t})}\Big] = H(q(c|\tau_{:t})) - H(q(c|\tau_{:t+1})). \tag{10}$$

This reward favorites actions which can reduce the entropy of cluster inference; and therefore, a trajectory leading to a consistent cluster inference is preferred. To more explicitly measure the divergence between the inferred cluster distributions from two steps, we define another reward encouraging consistent cluster inference,

$$r_c(a_t) = -\mathrm{KL}(q(c|\tau_{:t}) \,\|\, q(c|\tau_{:t+1})). \tag{11}$$

The above two rewards are designed to collect certain and consistent cluster-level information. After we locate the cluster, the exploration policy should focus on finding task-level information, which becomes easier under the help of cluster information. Intuitively, cluster is easier to find, as information across individual tasks can be leveraged. This induces a coarse-to-fine exploration strategy. We define the following composed reward to encourage the coarse-to-fine exploration behaviors,

$$r_e(a_t) = r(a_t) + \gamma_h(t)r_h(a_t) + \gamma_c(t)r_c(a_t), \tag{12}$$

where $r$ is the reward provided by the environment. $\gamma_h$ and $\gamma_c$ are two temporal decaying functions,

$$\gamma_h(t) = b_h - a_h \exp(-s_h(H - t)), \; \gamma_c(t) = -b_c + a_c \exp(-s_c(H - t)), \tag{13}$$

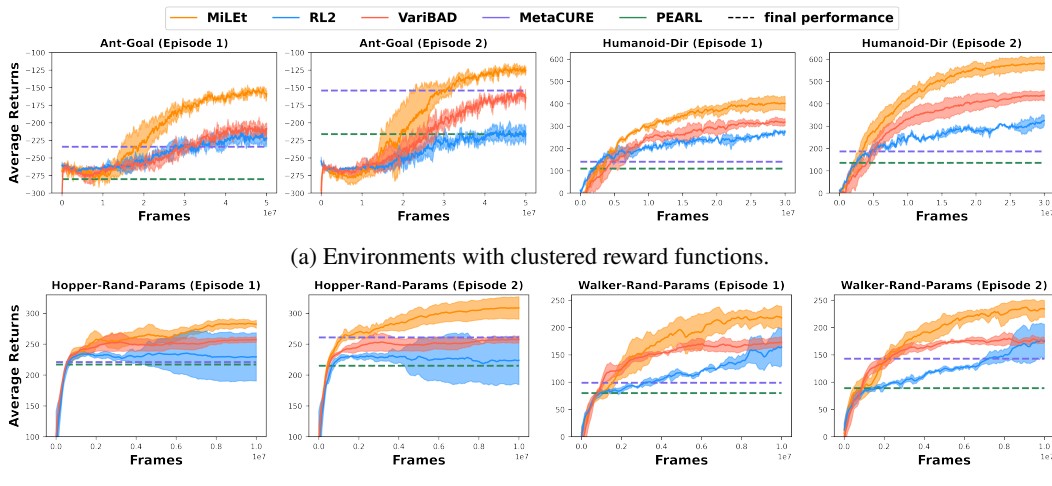

(a) Environments with clustered reward functions.

(b) Environments with clustered state transition functions.

Figure 2: Average test performance for 2 episodes on MuJoCo environments.

where $\{a, b, s\}_{h,c}$ are hyper-parameters controlling the rate of decay. $\gamma_h$ should gradually decrease to 0, which means we encourage the policy to find a certain cluster at the early stage, but it should begin to collect task information later. $\gamma_c$ gradually increases from a negative value to positive. At the early stage, a negative $\gamma_c$ encourages the policy to try different clusters. Later, a positive $\gamma_c$ enforces the policy to stick to the current cluster and focuses more on discovering task information by maximizing raw rewards.

Finally, the exploitation policy $\pi_\phi$ inherits the hidden state $h_H^{\pi_\psi}$, which encodes knowledge collected by the exploration policy, and is then trained to maximize the expected reward defined in Eq.(1). The detailed pseudo-codes of meta-train and meta-test phases for MILET are shown in Appendix B.

# 5 EXPERIMENTS

To fully demonstrate the effectiveness of MILET in handling structured heterogeneity in meta-RL. In this section, we conduct extensive experiments to study the following research questions: (1) Can MILET achieve better performance than state-of-the-art meta-RL algorithms by handling structured heterogeneity in the task distribution? (2) Can MILET effectively unravel cluster structures in both rewards and state dynamics? (3) Can the sparse reward issue be mitigated by exploratory clustering of tasks? (4) How does the number of clusters affect the final performance of MILET? We defer the ablation study to Appendix E.

## 5.1 REWARD AND DYNAMICS CLUSTER ENVIRONMENTS

**Environment setup.** We evaluate MILET on two continuous control tasks with *clustered reward functions*, simulated by MuJoCo (Todorov et al., 2012). In **Ant-Goal**, the ant robot needs to move to a predetermined goal position. We create 4 clusters by distributing the goal positions in 4 different centered areas. In **Humanoid-Dir**, the human-like robot is controlled to move towards different target directions. We create 4 clusters by distributing goal directions along 4 orthogonal directions in a 2D space. We also created environments with *clustered transition functions* by adopting two movement environments **Hopper-Rand-Params** and **Walker-Rand-Params**, also simulated by MuJoCo. The physical parameters of the robot, including *body mass*, *damping on degrees of freedom*, *body inertia* and *geometry friction*, can be manipulated to realize different transition functions of the robot's movement. The hopper and walker robots are required to move smoothly under different parameter settings. We created 4 clusters by manipulating one of the parameters at a time and keep the others to the default parameters. The detailed procedure of task generation can be found in Appendix C.

**Baseline setup.** We compared MILET with several representative meta-RL baselines, including RL[2] (Duan et al., 2016), PEARL (Rakelly et al., 2019), VariBAD (Zintgraf et al., 2020) and MetaCURE (Zhang et al., 2021). For each environment, we created 500 tasks for meta-train and hold out 32 new tasks for meta-test. We report the performance on test tasks over frames in the meta-train phase. In the meta-test phase, we executed 2 episodes in each new task. For algorithms

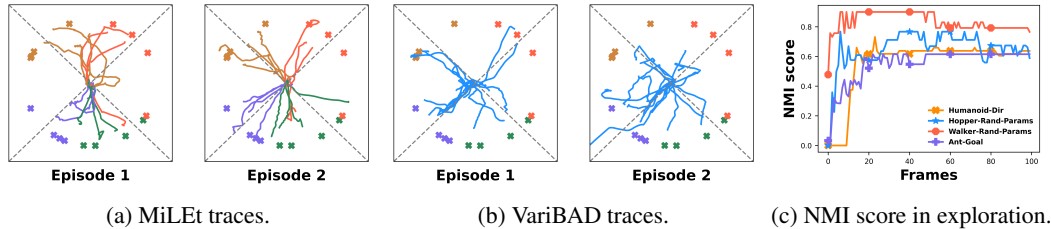

| (a) MiLEt traces. | (b) VariBAD traces. | (c) NMI score in exploration. |

Figure 3: Qualitative analysis of MILET. (a) Traces of MILET on the meta-test tasks of Ant-Goal. Cross marks represent goal positions, and the colors represent the cluster assignments produced by MILET. The dashed lines suggest the optimal traces to the centers of ground-truth clusters. (b) Traces of VariBAD on the same meta-test tasks of Ant-Goal. The traces are in the same color as VariBAD is unaware of clusters. (c) NMI of MILET's inferred clusters in the exploration episode of meta-test tasks in four environments.

with the explicit exploration policy, i.e., MILET and MetaCURE, we run their exploration policy in the first episode and exploitation policy in the second episode. We used public algorithm implementations provided by the original papers. We trained MILET via Proximal Policy Optimization (PPO) (Schulman et al., 2017) and set the default cluster number $C$ to 4. PEARL and MetaCURE are based on off-policy algorithms (Haarnoja et al., 2018); they need less frames of data to converge in the meta-train phase. We terminated them once the algorithm was converged and reported the final performance. The reported performance is averaged over 3 random seeds. More hyper-parameter settings and implementation details can be found in Appendix D.

**Results and analysis.** Figure 2 shows the test performance of all evaluated meta-RL algorithms. We also provide qualitative analysis in Figure 3, including visualization of the models' behaviors and the clustering performance of MILET in the exploration episode measured by the normalized mutual information score (NMI).

MILET showed significant improvement against baselines in the second episode, which is expected as it benefits from the information explored in the first episode. Interestingly, we can observe even though the first episode of MILET was reserved for exploration, it still performed comparable to other methods in all four different environment setups. In the first episode, MILET behaves exploratorily to find the most probable cluster of the current task, and thus its traces in Figure 7a look like spirals from the starting point. VariBAD is also designed to explore by uncertainty in task inference, but its traces are close to random walk at the early stage, which is less effective. In Figure 3c, we can observe the NMI scores of the MILET's inferred tasks have almost converged in 20 time steps, which means the stable cluster inference can already provide the agent necessary cluster information to gain fine-grained task information. This also explains how MILET obtained comparable performance in the first episode. In the second episode, with the better task information, MILET is able to move towards the targets directly, showing significant improvements against baselines. MetaCURE guides the exploration by task IDs, which in fact provides more information of environment than what MILET can access. However, the exploration empowered by task IDs focuses more on leveraging task-level information thus is less effective in exploring coarser but useful information, e.g., clusters, especially when the task IDs are independent on the task structure.

### 5.2 SPARSE REWARD ENVIRONMENTS

**Environment setup.** We evaluated MILET on a more challenging setting where the reward is sparse. We modified Ant-Goal and Humanoid-Dir environments such that the agent only gets positive rewards within a small region around target positions or target directions, otherwise 0. We denote them as **Ant-Goal-Partial** and **Humanoid-Dir-Sparse**. In Ant-Goal-Partial, we found all evaluated methods failed when rewards were too sparse. Hence, we set the reward regions to cover the initial position of the robot. With sparse reward, it becomes more important for the agent to leverage knowledge across related tasks, as the feedback within a single task is insufficient for policy learning. More environment details are deferred to Appendix C.

**Results and analysis.** We present results in Figure 4. By exploring and leveraging task structures in the exploration episode, MILET outperformed all baselines with a large margin. MetaCURE is designed for exploration when reward is sparse by utilizing task ids, which indeed helped it outperform other baselines in Ant-Goal-Partial. But such exploration is on the task-level thus is unable to effectively explore cluster information to enhance task modeling. On the contrary, MILET lever-

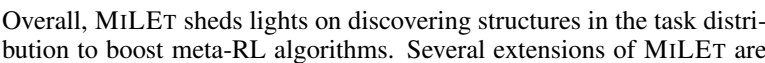

Figure 4: Average test performance for 2 episodes on sparse reward environments.

aged relatedness among tasks with in the same cluster to bootstrap policy adaptation; and as shown in Figure 3c, cluster inference can be efficiently solved, which provides an edge for accurate task inference. These two factors contribute to MILET's better performance in the exploitation episode.

### 5.3 INFLUENCE OF THE NUMBER OF CLUSTERS

In this experiment, we study how the number of clusters $C$ influence the final performance, especially when there is a mismatch between the ground-truth number of clusters and $C$ set by the agent. We set $C$ to different values and denote it in suffixes. We additionally create a set of tasks on the Ant-Goal environment, where the goal positions are uniformly sampled on a circle. We denote it as **Ant-U**. In this setting, there is no explicitly created clusters of tasks.

Table 1: Performance comparison on Ant-Goal and its variant.

|          | Ant-Goal      | Ant-U         |
|----------|---------------|---------------|
| VariBAD  | -168.6±9.6    | -162.4±9.2    |
| MILET-2  | -132.3±7.6    | -128.6±8.8    |
| MILET-4  | -125.4±5.1    | -113.7±4.8    |
| MILET-6  | -123.6±4.4    | -99.7±5.2     |
| MILET-8  | -124.2±4.7    | -117.9±5.7    |
| MILET-10 | -128.6±5.2    | -142.7±10.4   |

The average final returns are shown in Table 1. Interestingly, we observe MILET can perform well even though there are no explicit cluster structures in Ant-U. By looking into the detailed trajectories, we found MILET segmented the circle into different parts as shown in Figure 5 such that knowledge from nearby tasks can be effectively shared. VariBAD mistakenly assumed all tasks can share knowledge and thus failed seriously. When $C$ is set smaller than the ground-truth number of clusters, MILET-2 discovers more general structures. However, transferable knowledge within such structures is less, causing the performance drop. However, it does not mean more clusters than necessary is helpful, as less knowledge could be shared in each cluster. In Ant-U, MILET-8 and -10 generate unnecessary clusters, and cluster assignments are mixed at the boundary of adjacent clusters (see visualization in Appendix G). Such inaccurate cluster modeling causes ineffective exploration and sharing of wrong knowledge, leading to degenerated performance. Hence, correctly inferring the number of clusters is an important future work.

## 6 CONCLUSION & FUTURE WORK

In this paper, we present MILET, a clustering-based method to unravel structured heterogeneity of tasks in meta-RL. MILET is able to discover clustered task structures in a population of RL-tasks and adapt cluster-level transferable information to new tasks. To quickly identify the cluster assignment of new tasks, MILET learns a separate exploration policy which aims to reduce uncertainty in cluster inference. We further design a dedicated reward function to control the exploration between the cluster- and task-level information. MILET outperformed representative meta-RL methods by utilizing structured heterogeneity in the task distribution. We further demonstrated its exploration on cluster structures greatly improves its sample efficiency in sparse reward environments.

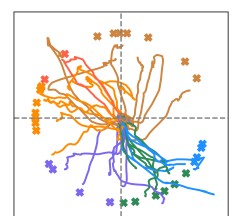

Figure 5: Traces of MILET-6 in exploration episodes on Ant-U. Colors represent the cluster assignments produced by MILET-6.

Overall, MILET sheds lights on discovering structures in the task distribution to boost meta-RL algorithms. Several extensions of MILET are worth to explore in the future work. We currently assume a uniform cluster prior. More complicated priors can be considered to enable new features. For example, the Gaussian process prior can be used to automatically identify number of clusters in the task distribution, and possibly detect out-of-distribution tasks in the meta-test phase. Also, MILET can be combined with skill-based RL methods (Pertsch et al., 2021; Nam et al., 2022) to learn cluster-level skills, which could form a new basis for meta-RL, e.g., each task can be modeled as a mixture of skills and different clusters of task associate with different skill distributions.

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

# A  ELBO DERIVATION

$$\mathbb{E}_{\rho(\mathcal{M},\tau^+)}\Big[\log p(\tau^+)\Big] = \mathbb{E}_\rho\Big[\ln \int p(z,c,\tau^+)\frac{q(z,c|\tau_{:t})}{q(z,c|\tau_{:t})}\Big]$$

$$= \mathbb{E}_\rho\Big[\ln \mathbb{E}_{q(z,c|\tau_{:t})}\Big[\frac{p(z,c,\tau^+)}{q(z,c|\tau_{:t})}\Big]\Big]$$

$$\geq \mathbb{E}_\rho\Big[\mathbb{E}_{q(z,c|\tau_{:t})}\Big[\ln \frac{p(z,c,\tau^+)}{q(z,c|\tau_{:t})}\Big]\Big]$$

$$= ELBO_t$$

where the inequality holds due to the Jensen's inequality. The inner expectation term can be further decomposed as,

$$\mathbb{E}_{q(z,c|\tau_{:t})}\Big[\ln \frac{p(z,c,\tau^+)}{q(z,c|\tau_{:t})}\Big]$$

$$= \mathbb{E}_{q(z,c|\tau_{:t})}\Big[\ln \frac{p(\tau^+|z,c)p(z|c)p(c)}{q(z|c,\tau_{:t})q(c|\tau_{:t})}\Big]$$

$$= \mathbb{E}_{q(z,c|\tau_{:t})}\Big[\ln p(\tau^+|z,c) + \ln \frac{p(z|c)}{q(z|c,\tau_{:t})} + \ln \frac{p(c)}{q(c|\tau_{:t})}\Big]$$

$$= \mathbb{E}_{q(z,c|\tau_{:t})}\Big[\ln p(\tau^+|z,c)\Big] - \mathbb{E}_{q(c|\tau_{:t})}\Big[\mathrm{KL}(q(z|c,\tau_{:t}) \parallel p(z|c))\Big] - \mathrm{KL}(q(c|\tau_{:t}) \parallel p(c)),$$

which is essentially Eq.(4). In practice, we first apply the Gumbel-Softmax trick (Jang et al., 2017) to sample a $\tilde{c}$ to calculate the second term instead of directly calculating the expectation, and then sample a $\tilde{z}$ given $\tilde{c}$ to calculate the first term using the reparameterization trick (Kingma & Welling, 2013). To calculate the last term, we decompose it into,

$$-\mathrm{KL}(q(c|\tau_{:t}) \parallel p(c)) = H\big[q(c|\tau_{:t})\big] + \mathbb{E}_{q(c|\tau_{:t})}\big[p(c)\big],$$

where $p(c)$ is the uniform prior of the cluster distribution, thus the last KL term is essentially the entropy of the posterior $q(c|\tau_{:t})$ plus a constant value, which can be easily computed.

# B  ALGORITHMS

---

**Algorithm 1:** MILET: Meta-train Phase

---

**Input:** A set of meta-train tasks $\mathcal{M}$ drawn from $p(\mathcal{M})$;
Initialize a buffer $\mathcal{V}$ for CBVI training;
**while** *not* Done **do**

    Sample a task $M_i \sim \mathcal{M}$;
    Collect exploration and exploitation episodes $\tau^+ = \{\tau^\psi, \tau^\phi\}$ by running Alg.(2) on $M_i$;
    Insert $\tau^+$ to $\mathcal{V}$;
    Compute intrinsic rewards $r_h$ and $r_c$;
    Train $\pi_\psi$ on $\tau^\psi$ by maximizing Eq.(12) using PPO;
    Train $\pi_\phi$ on $\tau^\phi$ by maximizing Eq.(1) using PPO;
    Sample a trajectory batch $v$ from $\mathcal{V}$;
    Update $\theta, \omega$ using $v$ by maximizing Eq.(8);

**end**

---

Algorithm 1 summarizes the training procedure of MILET. We use the on-policy algorithm PPO (Schulman et al., 2017) to train policies; and retain a buffer storing trajectories for the cluster-based variational inference training. Algorithm 2 is used to obtain trajectories given a task. When it is executed in Algorithm 1 during training, $c$ is sampled using the Gumbel-Softmax trick, otherwise using argmax when handing new tasks during testing.

---

**Algorithm 2:** MiLET: Meta-test Phase

---

**Input:** Meta-test task drawn from $p(\mathcal{M})$, number of exploitation episodes $N$;

**for** $t = 1, ..., H$ **do**

    Obtain $c = \arg\max_c q_\theta(c|h^\psi_{\alpha,t-1})$ ;     // Use Gumbel-Softmax during training

    Take action according to $\pi_\psi\big(a_t|s_t, q_\theta(z|h^\psi_{\beta,t-1}, c)\big)$;

    Update $h^\psi_t$ with $(s_t, a_t, r_t, s_{t+1})$;

**end**

Initialize $h^\phi_0 = h^\psi_H$;

**for** $e = 1, ..., N$ **do**

    **for** $t = 1, ..., H$ **do**

        Obtain $c = \arg\max_c q_\theta(c|h^\phi_{\alpha,t-1})$ ;  // Use Gumbel-Softmax during training

        Take action according to $\pi_\phi\big(a_t|s_t, q_\theta(z|h^\phi_{\beta,t-1}, c)\big)$;

        Update $h^\phi_t$ with $(s_t, a_t, r_t, s_{t+1})$;

    **end**

**end**

---

## C  ENVIRONMENTS

**Ant-Goal.** We create clusters by manipulating the goal positions. Given an angle $\theta \in [0, 2]$, we obtain the 2D coordinate of the goal position as $(r\cos\theta, r\sin\theta)$, where $r$ is the radius and fixed to 2 in our experiments. To create clusters, we sample $\theta$ from 4 different normal distributions $\mathcal{N}(0.25, 0.2^2)$, $\mathcal{N}(0.75, 0.2^2)$, $\mathcal{N}(1.25, 0.2^2)$, $\mathcal{N}(1.75, 0.2^2)$. To sample a task, we first uniformly sample a normal distribution, and then sample a $\theta$ from it.

**Humanoid-Dir.** We create clusters by manipulating the goal directions. To sample a goal direction, we first sample an angle $\theta \in [0, 2]$, and then the goal direction is $(\cos\theta, \sin\theta)$. To create clusters, we sample $\theta$ from 4 different normal distributions $\mathcal{N}(0.25, 0.2^2)$, $\mathcal{N}(0.75, 0.2^2)$, $\mathcal{N}(1.25, 0.2^2)$, $\mathcal{N}(1.75, 0.2^2)$. To sample a task, we first uniformly sample a normal distribution, and then sample a $\theta$ from it.

**Hopper-Rand-Params.** We create clusters by manipulating physical parameters of the robot, causing different transition functions. We have four sets of parameters in total, including *body mass*, *damping on degrees of freedom*, *body inertia* and *geometry friction*. To sample a task, we first uniformly sample one of the four parameter sets, and then multiply parameters in it with multipliers sampled from $\mathcal{N}(3, 1.5^2)$.

**Walker-Rand-Params.** The parameter sets are the same as the hopper robot. We find small multipliers cannot change transition functions a lot in this environment, thus we sample the multipliers from $\mathcal{N}(6, 1^2)$.

**Ant-Goal-Partial.** The tasks are sampled in the same way as Ant-Goal. The goal reward is defined as,

$$r = \begin{cases} -|x - g|_1 + t, & |x - g|_1 \le t \\ 0, & otherwise \end{cases}$$

where $x$ is the current coordinate of the robot, $g$ is the coordinate of the goal position, $t$ is the threshold, whish is fixed to 3 in our setting. Control cost and contact cost are also included in the final reward.

**Humanoid-Dir-Sparse.** The tasks are sampled in the same way as Humanoid-Dir. The velocity reward is defined as,

$$r = \begin{cases} g \cdot v, & \frac{g \cdot v}{\|g\|\|v\|} \ge t \\ 0, & otherwise \end{cases}$$

where $v$ is the velocity of the robot, $g$ is the goal direction, $t$ is the threshold and fixed to 0.8 in our setting. Control cost and contact cost are also included in the final reward.

**Ant-Goal-U.** For each task, we uniformly sample a $\theta \sim \mathcal{U}(0, 2)$ and obtain the goal position $(r\cos\theta, r\sin\theta)$ with $r$ fixed to 2.

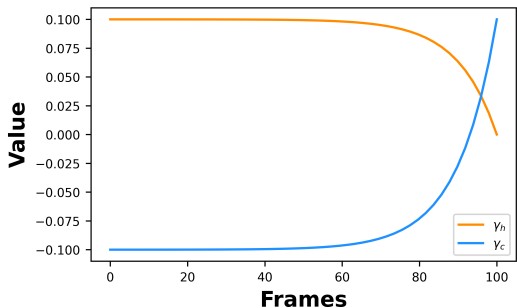

Figure 6: Visualization of $\gamma$ in the exploration policy.

## D    HYPER-PARAMETERS

We used the PyTorch framework Paszke et al. (2019) to implement our experiments. We use PPO with Humber loss to update MILET. In the cluster-based variation inference, we set $\lambda_I$ to 1, and $\lambda_P$ to 0.1. We update the target network $p_{tgt}(z|c)$ every 50 training epochs. For the decaying function used in exploration policy, we set $\gamma_h(t) = 0.1 - 0.1 \exp(-0.1(H - t))$ and $\gamma_c(t) = -0.1 + 0.2 \exp(-0.1(H - t))$, which are visualized in Figure 6. We use the Adam optimizer (Kingma & Ba, 2014) for both policy learning and cluster-based variational inference training. For policy learning including $\pi_\psi$ and $\pi_\phi$, we set the learning rate as 1e-4, while 1e-3 for variational inference training. In the clustered reward function environments, $\lambda_s$ is set to 0 in MiLEt and all variational inference baselines. In the clustered state transition function environments, $\lambda_s$ is set to be 1. The other parameters are set as default. The max episode length $H$ is set to 100 for every environment.

We provide implementation of MILET and task generation process in the supplementary material.

## E    ABLATION STUDY

We study the contribution of proposed components in MILET. Firstly, we disabled the exploration policy and directly train the exploitation policy to maximize task rewards. Secondly, we removed the stacked GRU (S-GRU) structure and only keep a single GRU to encode the trajectory $\tau_{:t}$ in the cluster-based variational inference. Lastly, we disable the consistency regularizers (CR) in the variational inference and directly optimize the ELBO. We use the same environments as Section 5.1.

The final performance of all variants is shown in Table 2. Firstly, without the exploration policy, the performance decreases significantly. This variant can still unravel clustering structures of tasks, so it performs better than VariBAD in general, which uses a simpler variational inference component. This again proves the necessity of handling structured heterogeneity in task distribution. However, it cannot effectively utilize local knowledge as the exploration does not aim to explore different cluster structures. Our exploration policy utilizes the cluster structure information and is trained to explore the most certain clusters for tasks, leading to better adaptation performance. Secondly, removing S-GRU makes it harder to utilize patterns with different levels of granularity in the interaction history. This ability is essential to quickly identify cluster structures as cluster patterns are more common across tasks, without which the exploration is less effective, causing performance drop. Also, a consistent variational inference is very important. Our designed consistency regularizers are helpful to discover cluster structures following the nature of sequential data in meta-RL, improving the final performance of MILET.

## F    VISUALIZATION OF STATE TRANSITION FUNCTIONS

We visualize the clusters in clustered transition function experiments in Figure 7. It is hard to directly visualize the state transition function as it is not a simple tabular function and it is conditioned on agent's actions. Recall that we created the ground-truth clusters by applying different multipliers on the initial parameters of each of the parameter sets, and thus such multipliers well characterize

Table 2: Ablation analysis of MILET.

|  | Ant-Goal | Humanoid-Dir | Hopper-Rand-Params | Walker-Rand-Params |
|---|---|---|---|---|
| VariBAD-G | -171.2±9.7 | 514.3±24.6 | 272.9±11.3 | 153.6±9.4 |
| ¬exploration | -130.4±4.5 | 533.9±18.4 | 278.3±9.6 | 164.2±16.7 |
| ¬S-GRU | -173.7±14.2 | 554.2±12.7 | 284.4±14.3 | 211.7±9.8 |
| ¬CR | -142.8±8.7 | 528.1±35.6 | 301.6±12.2 | 219.4±11.1 |
| MILET | -125.4±5.1 | 577.7±28.0 | 312.2±18.1 | 232.6±15.4 |

the state transition in each task. We then calculate the average multipliers of tasks in each inferred cluster [4]. Ideally, one cluster of tasks should only have one set of multipliers larger than 1, as it is how we created different task clusters. From this visualization results, we can clearly observe that MILET successfully identifies the difference of 4 clusters.

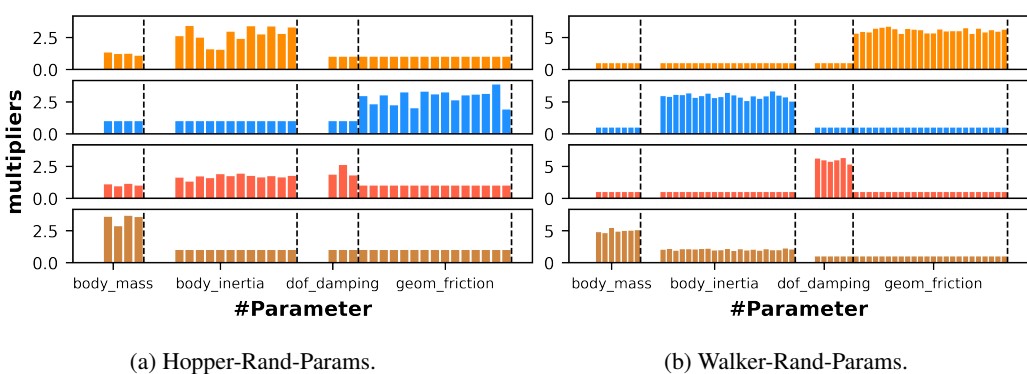

(a) Hopper-Rand-Params.

(b) Walker-Rand-Params.

Figure 7: Clusters in state transition functions. Each color represents an identified cluster by MILET. Four parameter sets are split by dashed lines.

## G  VISUALIZATION ON ANT-U

We visualize the task clusters identified by MILET-8 and MILET-10 on the testing tasks in the Ant-U environment. We can observe that tasks are split into smaller groups by these two new variants, and their cluster assignments are mixed at the boundary of adjacent clusters. Such inaccurate cluster modeling causes ineffective exploration and sharing of wrong knowledge across tasks, leading to the degeneration of final performance in Table 1.

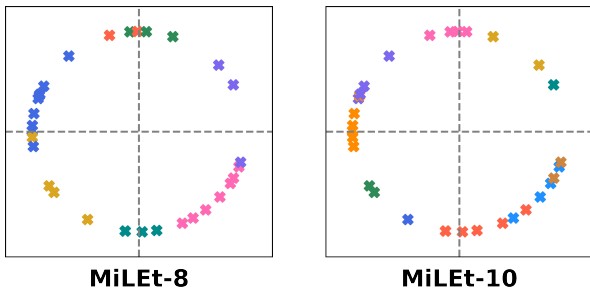

Figure 8: Cluster assignments produced by MILET-8 and MILET-10 in the Ant-U environment.

---

[4]Multipliers with value 0 mean the corresponding initial parameters are 0.

# H    DISCUSSION OF CLUSTER-AWARE EXPLORATION

Our exploration policy distinguishes MILET from other task inference methods in utilizing the structures of the task distribution. VariBAD (Zintgraf et al., 2020) performs exploration as a function of task uncertainty by maximizing task rewards. MetaCURE (Zhang et al., 2021) and DREAM (Liu et al., 2021) propose to explore by maximizing the mutual information between inferred task embeddings and pre-defined task descriptions. These exploration methods are unaware of structures of the task distribution, thus are less effective in exploring coarser but useful information, i.e., clusters. Our cluster-aware exploration is designed to quickly reduce the uncertainty in cluster inference. As shown in Figure 7a, the agent explores on the map scale to find the most suitable structure.

To better understand the implication of our cluster-aware exploration, we compare the clustering quality (measured by NMI score) at the end of the first episode of MILET and its variant without the exploration policy in Table 3. Similar to VariBAD, this variant performs exploration by considering task uncertainty. Our exploration policy with explicit clustering objective obtains better clustering quality, which builds foundation for refining task inference, resulting in better final performance in Table 2.

Table 3: NMI score of different exploration methods.

|              | Ant-Goal | Humanoid-Dir | Hopper-Rand-Params | Walker-Rand-Params |
|--------------|----------|--------------|--------------------|--------------------|
| ¬exploration | 0.466    | 0.485        | 0.517              | 0.327              |
| MILET        | 0.615    | 0.639        | 0.588              | 0.765              |

