# OpenReview forum: "Unravel Structured Heterogeneity of Tasks in Meta-Reinforcement Learning via Exploratory Clustering"
_ICLR.cc/2023/Conference — Submitted to ICLR 2023_

### Official Review · Reviewer_nQz2 · 2022-10-22

**Confidence:** 4
**Correctness:** 4
**Technical Novelty And Significance:** 3
**Empirical Novelty And Significance:** 3
**Recommendation:** 6

**Clarity, Quality, Novelty And Reproducibility:**

`Clarity & Quality:`
Overall good although several improvements could be made:
- Figure 1 appears relatively early in the text, its current form makes it difficult to parse. The information flow in the graphic is at first unclear and confusing. The Encoder/Decoder structure could be in a separate figure, as it can be seen more as a detailed view of the policy, rather than a separate component as the figure suggests. The interaction between Environments and Policies should be a circular one, rather than  information flowing only one way (as is shown at the moment). Quantities such as $\mathbf{r}_h$, $\mathbf{r}_c$ are unclear until the very end of the methodology section 4 - a label could possibly help, perhaps there is no need for it to be included. Similarly for the policy hidden states. In the encoder graphic, the authors should be clear and show that a sample from the cluster distribution is passed to estimate the within-cluster distribution (similarly to how sampling is explicit between encoder/decoder).
- Eq (3) and Eq (4) could simply make the notation for samples from the respective distributions clear, as the authors feel the need to explain it in the subsequent text currently.
- Eq (4): Please make the form of $q_{\theta}(c|\tau_{:t})$ clear and state that the Gumbel-softmax trick is being used
- After Equation (4): Could the authors explain why the L2 distance is used. What makes the log-likelihood intractable (as claimed)?
- Figure 2 & 4: Please fix the y-scale for experiments on the same task. Otherwise the relative difference is more difficult to observe.
- Please number all equations so they can be referred to easily.



`Novelty:`
- Sufficient

`Reproducibility:`
- Difficult to assess without a explicit attempt, although I am reassured by the fact that authors provide their implementation in the supp. material.


**Strength And Weaknesses:**

`Strengths:`
- Task clustering in Multi-task and Meta-RL is an important yet challenging problem and various attempts at designing such algorithms that make use of clustering have been unsuccessful. The authors correctly identified this as an important and timely problem.
- Clarity in the algorithmic derivation and motivation of the work (Several minor suggestions for improvement are provided).
- Results on the provided experiments are strong.

`Weaknesses:`
- The provided experiments are relatively simple and hand-designed, cluster information is relatively local and of limited abstraction. At this point I would be unsure whether the technique would be capable of clustering entire tasks from e.g. the MetaWorld task suite based on the more abstract problem being solved (e.g. various pushing/pull tasks, different tasks involving the same object (e.g. buttons, coffee mugs)).
- The method does not handle unknown number of clusters.

`Requests/Questions:`
- Could the authors kindly provide a cluster visualisation of the clustered transition function experiments?
- Table 1: At what point do increasing number of clusters reduce performance?

**Summary Of The Paper:**

This paper introduces a method for explicit task clustering as an inductive bias for Meta-RL algorithms operating on a task distribution. The key assumption is heterogeneity within that task distribution, i.e. that the i.i.d. assumption of draws from the task distribution is unrealistic in real Meta-RL setting and that knowledge can be more readily transferred by explicitly modelling shared structure. To that end, the authors propose a hierarchical probabilistic model, along with an inference procedure that first infers cluster ids and then the task within that cluster. In addition, the method further shows a training procedure distinguishing between an exploratory policy for the first and an exploiting policy for subsequent episodes and introduced various intrinsic rewards designed to lead to stable inference. The method shows promising results, although tasks are of limited complexity and clusters correspond to low-level information (Goal region).

**Summary Of The Review:**

Overall a good submission with solid methodological improvements, albeit with somewhat limited complexity in its experimental section. In its current stage it is difficult to asses whether the method along with various introduced tricks scales to more complex scenarios, making it difficult to estimate the likely significance of this work in inspiriting future approaches. I'd consider raising my score if the authors improve the work in that regard during the rebuttal. Nevertheless a good submission on a challenging problem that would not be out of place as an ICLR publication.

---

> ### Author Response · Authors · 2022-11-14
> **Response to Reviewer nQz2**
>
> Thanks for the detailed suggestions and acknowledgment of our contributions! We have modified the submission according to the reviewer's suggestions to improve its clarity and quality. Next, we address the reviewer's comments/requests in detail.
>
> ***
> [Q1] Could the authors kindly provide a cluster visualization of the clustered transition function experiments?
>
> [A1] We provided it in Appendix F of the updated submission. We visualized the environment parameters (we manipulated them to create clusters with different state transitions) of each predicted cluster by MILET. We can clearly observe that MILET successfully identifies 4 clusters with respect to the parameter sets, i.e., the transition functions. This shows MILET can effectively cluster the tasks to realize their relatedness.
>
> ***
> [Q2] Table 1: At what point do increasing number of clusters reduce performance?
>
> [A2] It depends on the underlying task distribution: it is a trade-off between the task clustering accuracy and the number of tasks in the same cluster where local knowledge can be shared. We added the results of MILET-8 and MILET-10 in Table 1. When explicit task clusters exist (e.g., Ant-Goal) and set more than enough clusters in MILET, the model can still identify ground-truth clusters, but assign tasks at the boundary of two adjacent ground-truth clusters to new clusters. Tasks in each resulting cluster can still correctly share local knowledge, though fails to fully leverage all transferable knowledge. As a result, empirically we found setting too many clusters does not significantly hurt the model’s final performance in such environments. However, in Ant-U where there are no explicit task clusters in ground-truth, the number of clusters becomes much more sensitive, as the model might assign incorrect clusters, as shown in our visualization of inferred clusters in Figure 5 and added Appendix G, where tasks located near the boundary between clusters got their cluster assignment interleaved.
>
> ***
> [Q3] The provided experiments are relatively simple and hand-designed, cluster information is relatively local and of limited abstraction.
>
> [A3]  Please first refer to our discussion in the general response Q2.
>
> We also investigated the MetaWorld suite before the submission, but found it is not suitable for studying the structured task heterogeneity problem in this work. MetaWorld includes 50 types of tasks (e.g, pick a place, pull an object) and each type of tasks have variations (e.g., various target positions). Different types of tasks can be seen as clusters, but all the methods included in our experiments can easily identify them as the initial states of different types of tasks are set to be obviously different. Hence, identifying task types cannot bring additional benefits to MILET against our baselines. Furthermore, as tasks of the same type are uniformly sampled in MetaWorld, there is no explicit task clustering structure. Hence, MILET can hardly benefit from task cluster modeling. In our experiments, we considered an environment consisting of 4 types of tasks in MetaWorld {plate slide, plate slide back, plate slide side, plate slide side back}, and MILET is only ~3% better than the best baseline in success rate. On the other hand, the same way for creating tasks as in our existing MuJoCo experiments can be applied to create task clusters in the MetaWorld environments (e.g., explicitly create task clusters). But since we have reported the results on MuJoCo, this set of experiments will not provide new information about the comparison between MILET and baselines. Based on the analysis and considerations we mentioned above, we decided not to include results from MetaWorld in the paper.

---

> > ### Comment · Reviewer_nQz2 · 2022-11-18
> > **Response to authors**
> >
> > Dear authors, thank you for the detailed response and providing additional experimental results. Great to see that correct clustering is visible in Appendix F. However, I hope the authors understand that both the decreasing performance when the number of clusters is misspecified and the remaining relative simplicity of the experiments (point taken on the MetaWorld suggestion) make it difficult to justify an increase in my score. Hence I will maintain my score

---

### Official Review · Reviewer_XnEo · 2022-10-25

**Confidence:** 3
**Correctness:** 3
**Technical Novelty And Significance:** 2
**Empirical Novelty And Significance:** 3
**Recommendation:** 5

**Clarity, Quality, Novelty And Reproducibility:**

Clarity

There are some clarity issues in the scoping of the proposed meta-RL setting, as described above. Otherwise, the writing for the methods and experiments is reasonably clear. The structure of the presentation follows standard norms.

Quality

The execution of the work seems sound. The experiments are commensurate in extent in comparison to recent meta-RL works, though notably there are no experiments with high-dimensional observations (e.g. images). The significance is also somewhat limited by the fact that the proposed method is only assessed on benchmarks of the authors' own creation, rather than one from prior works that exhibits the characteristics motivating this work.

Novelty

The main novelty lies in proposing and studying the specific problem setting. The technical novelty is rather low as explained above.

Reproducibility

Chances of reproducibility are high since descriptions of the algorithm and environments are provided, as is code.


**Strength And Weaknesses:**

Strengths

The authors study a hitherto neglected property of realistic task distributions. The proposed method that leverages this property is sensible and includes new terms that address the specific challenges arising from inferring mixture ids. The empirical evaluation involved modifying existing benchmark tasks to reflect this property.

Weaknesses

The newly proposed meta-RL setting is not clearly distinguished from that of prior work. The authors mention that prior meta-RL assumes tasks are IID, but that seems to need to hold for this work as well. Assuming latent variables underlying the task distribution is well and good, but does the marginal task distribution need to exhibit clusters, or can it be relatively uniform? Put another way, If MILET can improve over prior methods for densely sampled task distributions, then perhaps the real benefit of MILET comes from leveraging the sufficiency of nearby tasks for transfer (a local property), as opposed to heterogeneity (a global property)? The novelty of the proposed method is limited as its backbone is essentially VariBAD (for the evidence lower bound) and MetaCURE/DREAM (for decoupled exploration and exploitation) endowed with the mixture task distribution assumption.

Minor

- I'm not a fan of the cooking analogy in the introduction. There is probably a lot of commonality in cooking Chinese and Indian dishes, which undercuts the point you're trying to make.

**Summary Of The Paper:**

This work proposes a method, MILET, that leverages an assumption of heterogeneity within the task distribution. More specifically, the task distribution is modelled as a mixture distribution. The objective of MILET essentially adapts VariBAD (Zintgraf et al., 2020), decomposing the belief as a mixture id and a task latent conditioned on the mixture id. The sampling of beliefs for the task decoder is restricted to the inferred posterior mixture id, operationalizing the intuition of eliminating the transfer between unrelated tasks. Two additional regularization terms are also proposed. MILET also separately maintains exploration and exploitation agents, similar to DREAM (Liu et al., 2021). Experiments are run on MuJoCo-based meta-RL domains in which the task variation manifests in the reward function (2 domains) or dynamics (2 domains). Overall, MILET is shown to outperform meta-RL methods from prior works that do not assume a mixture task distribution.

**Summary Of The Review:**

I currently recommend borderline reject due to the issues of clarity regarding the problem statement, limited technical novelty, and moderate empirical significance.

---

> ### Author Response · Authors · 2022-11-14
> **Response to Reviewer XnEo**
>
> Thanks for acknowledging the importance of the problem we are solving in this paper! We address the reviewer's comments/concerns as follows.
>
> ***
> [Q1] Motivation
>
> [A1] We agree that tasks in our setting are also IID sampled, but they are sampled from a multi-modal distribution, which is referred to as the structured heterogeneity in task distribution in our work. On the contrary, prior works mostly (implicitly) assumed a unimodal task distribution or sampled from a uniform distribution. We have revised the manuscript to make the problem statement clearer. The reviewer’s understanding of leveraging the sufficiency of nearby tasks is accurate about our solution. The structured heterogeneity induces local structures that can be utilized to enhance meta-RL, which is the key insight we leverage to solve the problem in this paper.
>
> ***
> [Q2] Novelty of our contribution
>
> [A2] The novelty of our work comes from handling a more realistic meta-RL setting, where the task distribution is multimodal. By modeling the clustering structure of tasks, we introduce a new exploration objective to reduce uncertainty in cluster inference. Compared with VariBAD, MILET is able to not only handle a mixture of task distributions, but also explore more effectively by having a dedicated exploration policy aiding cluster inference. Compared with DREAM and MetaCURE, MILET lifts the dependence on given task descriptions to learn the exploration policy by utilizing our newly proposed clustering objective.
>
> ***
> [Q3] Motivation example
>
> [A3] Thanks for pointing out that our cooking analogy does not well characterize the nature of the problem studied in this paper, as cooking Chinese and Indian dishes indeed shares a lot of commonalities. In the updated submission, we explained our problem scenario using the comparison between Go and Gomoku games, where the state space is almost the same, but the needed skills are obviously different since the rules are different.
>
> ***
> [Q4] The proposed method is only assessed on benchmarks of the authors' own creation, rather than one from prior works that exhibits the characteristics motivating this work.
>
> [A4] Please refer to our discussion in the general response Q2.

---

> > ### Comment · Reviewer_XnEo · 2022-11-18
> > **Review update**
> >
> > Thank you for taking the time to respond. I appreciate the clarifications. In light of the concerns of other reviewers (cxKJ, XdEV), I choose to retain my score.

---

### Official Review · Reviewer_cxKJ · 2022-10-25

**Confidence:** 4
**Correctness:** 2
**Technical Novelty And Significance:** 3
**Empirical Novelty And Significance:** Not applicable
**Recommendation:** 5

**Clarity, Quality, Novelty And Reproducibility:**

The paper is clearly written and suggests a novel algorithm.

As outlined above, my primary concern is with the interpretation of the results, which I am not convinced that they are correct.

**Strength And Weaknesses:**

# Strenghts

* The paper and problem setting is well motivated.
* The related work section is well written, but could be more extensive.
* The proposed ideas are interesting.

I personally find especially the way the exploration policy is learned quite interesting. For one, splitting exploration and exploitation policy might be a useful inductive bias in itself as it might provide a simple but effective way of balancing both aspects -- as opposed to trying to do reward engineering to learn one policy fulfilling both roles.
Secondly, the cluster entropy reward might be a very effective proxy reward for exploration success.

# Weaknesses

While I like the algorithm and can believe that it outperforms the baseline on the evaluated tasks, I am unsure about some of the justifications made for design choices and explanations surrounding the experimental results. I will discuss several instances of this below.

## Where does the advantage come from: latent hierarchy or exploration policy?

Unfortunately, the paper does very little to disentangle whether the success is coming from the clustering of the latent space or the separation between exploration and exploitation policy. Instead, it attributes most of its success to the clustering in latent space, claiming that this provides advantages in terms of knowledge sharing between tasks - which I'm not yet convinced by. In particular, I am not sure why VariBAD or Pearl would fail at effectively sharing that knowledge as their formulation seems general enough to do so.
And while MILET provides a hierarchical inductive bias in the latent space, the success of non-hierarchical VAEs has, imo, shown that these types of flat latent spaces are usually flexible enough.

## Consistency

In section 4.1 the paper discusses a problem with "consistency". However, it is nowhere defined what exactly is meant by consistency. In particular, I am not sure how this can be a contradiction to the update to the agents belief -- which should become more certain over time.

I assume that the authors mean by "non-consistent" if q(c|tau) 'jumps' from one time-step to the next in a way that the two KLs have very little overlap -- but either way, this should be made precise.

## Experimental Results

I find it quite surprising that MILET's _exploration_ policy outperforms the other methods by such a large margin -- this seems to me to be more concistent with the hypothesis that the split between exploration and exploitation policy is the real innnovation here and not the GMM in the latent space?

For the sparse reward setup, I am not sure I understand the following sentence:

> With sparse reward, it becomes more important for the agent to leverage knowledge across related tasks, as the feedback within a single task is insufficient for policy learning.

Given that the reward it sparse, how can knowledge sharing help: Before we find the goal, we cannot know which cluster it is in and after we find the goal, this becomes irrelevant as we've already found the goal. Again, imo this could be very well explained by the superior exploration policy and not the latent structure.

**Summary Of The Paper:**

The authors propose an inference based meta-learning approach, similar to Pearl or Varibad, but with a hierarchical latent task embedding modelled as GMM. Additionally, they propose to learn two policies, one optimised for exploration, one for exploitation. The algorithm is evaluated on MuJoCo tasks with variable goal locations which are sampled from clusters. They refer to this as "structured heterogeneity".

**Summary Of The Review:**

The paper proposes an interesting algorithm which seems to perform well in practise on tasks with "structured heterogeneity".
My main concern is with the discussion of the results and the justification of the algorithm. In particular, I am not convinced that the superior performance is due to the latent structure (as they authors claim). Additional discussion and, in particular, ablation studies would hence strenghten the paper.

---

> ### Author Response · Authors · 2022-11-14
> **Response to Reviewer cxKJ**
>
> Thanks for the acknowledgment of our idea! We address the reviewer's comments/concerns in detail as follows.
>
> ***
> [Q1] Where does the advantage come from: latent hierarchy or exploration policy?
>
> [A1] Please refer to our discussion in the general response Q1.
> ***
> [Q2] Consistency
>
> [A2] The in-trial consistency is defined as whether the same cluster assignment is obtained inside a trial. To enforce it, we penalize the inconsistency among the cluster assignments between any two consecutive time steps, which is defined as the KL divergence between two consecutive cluster posterior distributions in Eq.(5). We further explained our notion of consistency in the updated submission.
> ***
> [Q3] Experimental results
>
> [A3] As shown in Figure 3(c), we can see that cluster identification is almost converged in 20 steps in the exploration episode. Then the exploration policy begins to refine task inference by collecting task-specific rewards, under the help of cluster information. This is the desired behavior of a good exploration policy for task inference in such an environment. Thus it is expected that MILET’s exploration policy also outperforms the baselines.
> ***
> [Q4] Explanation of the sparse reward setting
>
> [A4] In this setting, the reward region of each task is small (i.e., actions taken outside this region do not receive any reward). The agent needs more effective exploration to identify the cluster, and then uses the cluster information to conduct specific task inference. For example in a locomotion task, after inferring that the goal is in the first quadrant, the agent needs to effectively utilize the knowledge of other tasks in this cluster, i.e., it needs to move towards the first quadrant, where it is more likely to collect more task-specific rewards to correctly identify the goal. Our exploration policy is trained to reduce the uncertainty in cluster inference, which provides the path for correct task inference.

---

### Official Review · Reviewer_XdEV · 2022-10-25

**Confidence:** 4
**Correctness:** 2
**Technical Novelty And Significance:** 3
**Empirical Novelty And Significance:** 3
**Recommendation:** 5

**Clarity, Quality, Novelty And Reproducibility:**

Generally, the clarity of this work is quite high. One aspect that is somewhat unclear is how the clusters c are used. From the presentation, MILET appears to never need the ground truth values of the clusters c, and only relies on its own predicted clusterings. However, the submitted code does appear to use the ground-truth values. Perhaps this is only for visualization purposes; I was unable to tell. I would appreciate some author clarification here, though.

The idea of clustering as an exploration objective is quite interesting and original.

**Strength And Weaknesses:**

 ## Strengths

- The idea to create an exploration objective based on identifying clusters of related tasks is interesting and completely novel.
- The experimental results are strong, although the experimental setup is imperfect, as I comment upon below.
- By and large, the presentation is quite clear and sufficiently technically precise.

## Weaknesses

A major weakness of this work is that the rhetorical claims and explanations are often unsupported, and in several places, clearly false. Many claims appear to be speculation, while they seem to be presented as fact. Below, I include several examples:
- This includes the primary motivation of this work: i.e., that tasks may be heterogeneous, where only some tasks may contain transferable information for other tasks, so it is crucial to identify clusters of transferable information. While identifying these clusters clearly holds some value, it is somewhat misleading, as identifying such clusters is _not_ necessary (i.e., a Bayes-adaptive optimal policy may achieve optimal returns, without any explicit knowledge of such clusters). Claims in the experiments like "such exploration is not effective enough since it is not able to recover underlying clustered task structures" are therefore misleading. Further, such claims are not even supported by the experiments themselves: results in the Appendix, which show that ablating the exploration policy from MILET continues to outperform VariBAD raise questions about what the primary source of empirical improvements is. As MILET outperforms baselines without even exploring to identify clusters, it's unclear if the gains are just from superior neural network architectures or not -- notably the baselines do not use the same architectures as MILET, and sharing the same architecture could help improve the experiments. Experiments on Ant-U without heterogeneous clusters, which are quite interesting, also suggest that it's not about finding transferability between tasks, and clustering can simply act as a useful auxiliary task / exploration objective. Finally, the claim that some tasks contain no transferable information to other tasks does not even appear to be true in the evaluated tasks or provided examples. In the example where clusters might be goals toward the top or the bottom, there are still shared motor skills. To be clear, I think the idea of clustering is still quite interesting, but proposing a story that is not supported is concerning.
- The experiments also speculate about the cause of reduced performance from MetaCURE, but present this as fact: "as in our setting all test tasks are new, the task id becomes meaningless in meta test. MetaCURE thus fails to generalize." MetaCURE does not use the task ID at meta-test time, so this is false.

It should be noted that exploration based on clustering can result in failure modes and is not always applicable. For example, in a distribution of tasks, where tasks can be clearly clustered into task-irrelevant but highly prominent features can lead to unhelpful exploration. In this case, the proposed reward bonuses may mislead exploration from finding actually useful information. Inclusion of such discussion could improve this work. As a related note, the tasks from the experiments do not in general require significant exploration, as much can be inferred from a single timestep reward from relatively random actions (although the exact reward details are not included for many of the tasks). This makes it somewhat challenging to evaluate the proposed exploration bonuses.

## Minor Comments
- Calling Ant-Goal-Sparse a "sparse" reward environment seems misleading if the rewards kick in even from the initial state and are not binary.
- Why are there only 5 colors in Figure 5, if it is MILET-6?
- Is the decomposition of the entropy H(q(c | tau)) into a telescoping series to create per-timestep rewards based on the similar decomposition from DREAM?



**Summary Of The Paper:**

This work proposes a new exploration objective in meta-reinforcement learning to discover clusters of related tasks. The claim is that tasks may be heterogeneous, where knowledge may only be transferable to other tasks within the same cluster, so exploring to identify these clusters could find the useful transferable information.

**Summary Of The Review:**

Overall, I generally like the proposed method ideas. Exploration via clustering is original and can clearly be helpful. However, the framing and rhetoric of this work is a severe weakness, so I currently cannot recommend acceptance. I look forward to the author response.

---

> ### Author Response · Authors · 2022-11-14
> **Response to Reviewer XdEV (1/2)**
>
> We appreciate the reviewer's acknowledgment of the novelty of our idea and clear presentation! We provide detailed responses to address the reviewer's concerns/comments.
>
> ***
> [Q1] A Bayes-adaptive optimal policy may achieve optimal returns, without any explicit knowledge of such clusters
>
> [A1] It is arguably true that the Bayes-adaptive optimal policy does not need explicit knowledge of task clusters; but finding the Bayes-optimal policies relies on accurate belief inference. For example, VariBAD performs posterior inference on task beliefs. However, VariBAD assumes a unit Gaussian prior, which can cause inaccurate task inference due to prior mismatch. We verified in Figure 2 that baselines designed under the unimodal task distribution assumption failed (e.g., VariBAD, MetaCURE) when the task distribution is multimodal. We have expanded our explanation of the performance of baselines in the updated submission to emphasize the impact of inaccurate belief inference.
> ***
> [Q2] As MILET outperforms baselines without even exploring to identify clusters, it's unclear if the gains are just from superior neural network architectures or not
>
> [A2] In Appendix E of the updated submission, we provide the result of another variant of MILET where we ablate the exploration policy and the S-GRU structure, and denote it as VariBAD-G. The neural network architecture of the actor-critic module and encoder-decoder network are the same as in VariBAD, but the VAE part is replaced by the Gaussian mixture VAE (GMVAE) to model the clustering structures of tasks. VariBAD-G also outperformed the baselines in general, showing that modeling the cluster structures is the most important, and our performance gain is not simply from the superior neural network architectures.
> ***
> [Q3] The claim that some tasks contain no transferable information to other tasks does not even appear to be true in the evaluated tasks or provided examples.
>
> [A3] We should emphasize that in this paper we do not assume the knowledge in different clusters is exclusive, and thus each cluster can not only contain overlapping global knowledge, i.e., motor skills, but also cluster-level knowledge, such as going toward the top or the bottom in Ant-Goal, which is not shared across clusters of tasks. We agree with the reviewer that explicitly modeling the global transferable and local knowledge is a promising direction, especially for handling OOD tasks. However, it is beyond the scope of our current work, and we leave it as an important future work.
> ***
> [Q4] In a distribution of tasks, where tasks can be clearly clustered into task-irrelevant but highly prominent features can lead to unhelpful exploration.
>
> [A4] We followed the standard meta-RL definition that tasks differ in reward and state transition functions. We agree that there are tasks that can be clustered by task-irrelevant features, for example, different map colors in the Ant environment. However, the essence of MILET is to cluster the tasks with respect to their optimal policies by modeling task relatedness. The task-irrelevant features will not change the reward/state transition functions, and thus they will not affect the optimal policies of tasks. Hence, they will not be captured by MILET. But we totally agree that inaccurate cluster modeling will induce unhelpful exploration and eventually hurt the model's performance. As we discussed in general response Q1, inaccurate cluster modeling causes ineffective exploration with large C on Ant-U, leading to the degeneration of final performance. This result further suggests that correct task inference is a prerequisite to obtaining good meta-RL performance in our problem setting.
> ***
> [Q5] Explanation of MetaCURE
>
> [A5] Thanks for pointing out our inaccurate description of MetaCURE. In the updated submission, we revised our explanation of MetaCURE to elaborate its exploration empowered by task ID focusing more on exploring task-level information thus less effective in exploring coarser but useful information about the relationship among tasks, such as clusters, especially when the task IDs are independently created from the task structure.

---

> > ### Author Response · Authors · 2022-11-14
> > **Response to Reviewer XdEV (2/2)**
> >
> > [Q6] One aspect that is somewhat unclear is how the clusters c are used
> >
> > [A6] Thanks for carefully reviewing our implementation! We provided ground-truth clusters only for debugging purposes. In our design, we only need to set the number of clusters, and the clusters can be automatically identified. MILET only uses its own estimated clusters.
> > ***
> > [Q7] Calling Ant-Goal-Sparse a "sparse" reward environment seems misleading
> >
> > [A7] We agree that Ant-Goal-Sparse is not a commonly defined sparse reward environment. But in this environment, the agent will not receive rewards when it moves out of the reward region, i.e., not every action leads to an immediate reward. We created this environment to show that MILET can still explore effectively in this harder setting by leveraging relationships among tasks. We renamed it as Ant-Goal-Partial in the revised submission to avoid confusion.
> > ***
> > [Q8] Why are there only 5 colors in Figure 5, if it is MILET-6?
> >
> > [A8] In Figure 5, we present the predicted clusters of the hold-out testing tasks, which are randomly sampled. There are only two tasks belonging to the red color (in the upper-right part of the figure) near the orange tasks, which might be less visible compared to other colors. Hence, there are indeed 6 colors in Figure 5.
> > ***
> > [Q9] Is the decomposition of the entropy H(q(c | tau)) into a telescoping series to create per-timestep rewards based on the similar decomposition from DREAM?
> >
> > [A9] Yes, similar to DREAM and MetaCURE, we decompose H(q(c | tau)) into telescoping series to create per-timestep rewards. The difference is we propose a new objective for exploration, i.e., reducing uncertainty in cluster inference, which gets rid of the dependence on given task descriptions, e.g., task IDs in DREAM and MetaCURE.

---

> > > ### Comment · Reviewer_XdEV · 2022-11-18
> > > **Reviewer Response**
> > >
> > > I appreciate the authors' response and clarification, which has helped alleviate my concerns about whether the claims in this work are well-supported, though some of my concerns remain, which I touch upon below.
> > >
> > > > [A1] It is arguably true that the Bayes-adaptive optimal policy does not need explicit knowledge of task clusters; but finding the Bayes-optimal policies relies on accurate belief inference. For example, VariBAD performs posterior inference on task beliefs. However, VariBAD assumes a unit Gaussian prior, which can cause inaccurate task inference due to prior mismatch. We verified in Figure 2 that baselines designed under the unimodal task distribution assumption failed (e.g., VariBAD, MetaCURE) when the task distribution is multimodal. We have expanded our explanation of the performance of baselines in the updated submission to emphasize the impact of inaccurate belief inference.
> > >
> > > I agree that the choice of prior can affect downstream performance, but this seems to be somewhat orthogonal to the original concern, which is that the Bayes-adaptive optimal policy need not contain explicit knowledge of task clusters. Hence, the wording should be careful to be about the choice of prior, rather than whether approximately solving a BAMDP is an appropriate method.
> > >
> > > **A3**. In light of this, it seems appropriate to tighten the wording in the introduction and throughout the work about the transferability of information between the clusters, then.
> > >
> > > > [A4] We followed the standard meta-RL definition that tasks differ in reward and state transition functions. We agree that there are tasks that can be clustered by task-irrelevant features, for example, different map colors in the Ant environment. However, the essence of MILET is to cluster the tasks with respect to their optimal policies by modeling task relatedness. The task-irrelevant features will not change the reward/state transition functions, and thus they will not affect the optimal policies of tasks.
> > >
> > > Task-irrelevant features such as changing the colors of things can indeed be reflected in the state transition function, which would in fact change the optimal policy.
> > >
> > > **A9**. If the technique for telescoping was used from prior works, it may be appropriate to appropriately cite.
> > >
> > > > All these results suggest that correct task modeling and inference is a prerequisite to obtain good meta-RL performance in our studied problem, while our cluster-aware exploration is the key step to achieve correct task inference.
> > >
> > > This seems promising, although it's unclear what the implications are for cluster-aware exploration. Task inference can be achieved via other methods, e.g., VariBAD, so a prevailing concern is what cluster-aware exploration buys you over other such methods. Perhaps it simply empirically performs well? It would be nice to discuss this thoroughly, even if a definitive answer cannot be reached.
> > >
> > > In general, I find the author response to be encouraging. However, my main concern about this work was to ensure that the claims are well-supported and cleanly demarcated from speculation, which I believe could require more careful revision, particularly around the introduction and framing of the benefits of task clustering. The author response itself could also, in some places, be improved in this regard, as highlighted above.
> > >
> > > I also acknowledge that my response is posted at the end of the discussion period, and the authors may not have the time to respond.

---

> > > > ### Author Response · Authors · 2022-11-19
> > > > **Author Response (1/2)**
> > > >
> > > > Thanks for the reviewer’s further comments and suggestions! We would like to address the remaining concerns of the reviewer as follows.
> > > > ***
> > > > [Q1] Bayes-adaptive optimal policy need not contain explicit knowledge of task clusters
> > > >
> > > > [A1] In the response, we pointed out that prior mismatch causes inaccurate belief inference, which barriers the finding of Bayes-optimal policies. In other words, fewer assumptions about the environment do not necessarily lead to satisfactory performance. In this paper, we focus on the setting where the underlying task distributions are clustered, and knowledge about optimal policy is only locally-transferable within clusters. Accordingly, we choose the Gaussian mixture prior to model the clusters in the task distributions, leading to better experimental performance in our extensive evaluations.
> > > > ***
> > > > [Q2] It seems appropriate to tighten the wording in the introduction and throughout the work about the transferability of information between the clusters.
> > > >
> > > > [A2] We have revised the draft to clarify that knowledge about the optimal policy for specific tasks in different clusters is not necessarily exclusive, and it is possible that different clusters share overlapping global knowledge. However, we want to emphasize the key point of this paper is that we explicitly model the clustering structures in the multimodal task distributions to capture the locally-transferable cluster-level knowledge about the policies. And our experimental results show that our method outperformed baselines that ignore the clustering structure, which supports our claim.
> > > > ***
> > > > [Q3] Task-irrelevant features such as changing the colors of things can indeed be reflected in the state transition function, which would in fact change the optimal policy.
> > > >
> > > > [A3]  Even if such features are included in the states or the state transition function, they should not change the optimal policy as they are “task-irrelevant”. More specifically, if such features are known to be irrelevant to the task, as the reviewer suggested, they should not affect how an optimal policy should behave, and thus not affect our algorithm that essentially clusters the optimal policies. Otherwise, they cannot be called “task-irrelevant” features.
> > > >
> > > > We guess the reviewer actually questions whether such features would affect task inference in practice. We agree that it is possible, especially when we do not have sufficient observations about the ongoing task. However, note that in Equation 5, we explicitly modeled both reward and state transition functions for clustered task modeling. Hence, even if the task-irrelevant features could cause variance when modeling the state transition functions, the reward function can provide important information for task inference. Again, if such features are “task-irrelevant”, they should not affect the behavior of the optimal policy, which is determined by both reward and state transition functions. For example, in our experiments, we follow previous works (e.g., VariBAD / MetaCURE / DREAM ) to set a hyper-parameter lambda_s to control the effect of the state transition function in variational inference. In the clustered reward function environments, lambda_s is set to 0 in MiLEt and all baselines. In the clustered state transition function environments, lambda_s is set to 1. We also explain it in the latest revision.

---

> > > > > ### Author Response · Authors · 2022-11-19
> > > > > **Author Response (2/2)**
> > > > >
> > > > > [Q4] If the technique for telescoping was used from prior works, it may be appropriate to appropriately cite.
> > > > >
> > > > > [A4] We have added the citations in the latest revision when we define the exploration reward.
> > > > > ***
> > > > > [Q5] It's unclear what the implications are for cluster-aware exploration. It would be nice to discuss this thoroughly, even if a definitive answer cannot be reached.
> > > > >
> > > > > [A5] We updated the thorough analysis of our exploration policy in Appendix H.
> > > > >
> > > > > Our exploration policy is different from other task inference methods in utilizing the structures of the task distribution. VariBAD performs exploration as a function of task inference uncertainty by maximizing task rewards. MetaCURE and DREAM propose to explore by maximizing the mutual information between inferred task embeddings and pre-defined task descriptions. These exploration methods are unaware of clustering structures in the task distributions, thus are less effective in exploring coarser but useful information, i.e., clusters. Our cluster-aware exploration is designed to quickly reduce the uncertainty in cluster inference, so as to quickly zoom into fine-grained task-level inference. As shown in Figure 7(a), the agent explores on the map scale to find the most suitable structure.
> > > > >
> > > > > To better understand the implications of our cluster-aware exploration, we further compare the clustering quality (NMI score) at the end of the first episode of MiLEt and its variant without the exploration policy in the newly-added Table 3. Similar to VariBAD, this variant performs exploration by only considering task uncertainty. The results show our exploration policy with explicit clustering objective obtains better clustering quality on all environments, which builds foundation for refining task inference, which is a piece of strong evidence for better final performance.

---

### Author Response · Authors · 2022-11-14
**General response to all reviewers**

We thank all reviewers for their insightful comments and suggestions! We have uploaded a revised version of our submission where the major changes are highlighted in blue. In the following, we will first respond to the common questions from all reviewers and then respond to each reviewer individually.
***
[Q1] Importance of latent task hierarchy inference vs. exploration policy

[A1] In the updated Appendix E, we include a new variant of VariBAD ablating its exploration policy and S-GRU, named VariBAD-G. Compared with the original VariBAD, we replace VAE with Gaussian mixture VAE (GMVAE) in this variant, which provides VariBAD the ability to model the latent task hierarchy. Its performance is better than most baselines. After adding S-GRU to enhance the latent task hierarchy modeling, the performance is further improved, which demonstrates that modeling latent task hierarchy plays a very important role in improving task performance.

Furthermore, we consider an Ant-Goal environment where we only keep one of the task clusters in Ant-Goal, and still train MILET with $C$=4. VariBAD obtained a cumulative reward of -121 while MILET got -118 in this setting. Hence, the unimodal prior in VariBAD can well handle this setting, because it is a correctly specified prior. The exploration policy of MILET did not help obtain a significant improvement as the clustering of tasks does not hold in this setting. We also provide results of MILET-8 and MILET-10 in Ant-U in the updated Table1. We found that with a larger C, the sampled tasks are split into smaller groups in these two new variants and cluster assignments are mixed at the boundary of two adjacent clusters (see our visualization in Appendix G). Such inaccurate cluster modeling causes ineffective exploration and sharing of wrong knowledge, leading to the degeneration of final performance.

All these results suggest that correct task modeling and inference is a prerequisite to obtain good meta-RL performance in our studied problem, while our cluster-aware exploration is the key step to achieve correct task inference.
***
[Q2] Benchmark setting

[A2] Current existing public meta-RL benchmarks do not consider the structured heterogeneity of tasks, i.e., they assume tasks are sampled from a unimodal distribution or completely uniformly sampled. We follow [1], where they manually combined several supervised meta-learning datasets to study the structured meta-learning problem, to create benchmarks suitable for studying the structured heterogeneity problem in meta-RL.

[1] Yao, Huaxiu, et al. "Hierarchically structured meta-learning." International Conference on Machine Learning. PMLR, 2019.

---

### Author Response · Authors · 2022-11-17
**A gentle reminder to the reviewers**

Dear reviewers,

Thank you again for your valuable time and thoughtful comments! We have provided detailed responses and additional experiment results to best answer the questions. As we are approaching the end of the discussion stage, we would appreciate it if you could read our responses and let us know if your concerns have been addressed. We are more than happy to further discuss any details that you find not fully addressed. Thank you.

Best,

Paper2050 Authors

---

### Decision · Program_Chairs · 2023-01-20

**Decision:**

Reject

**Justification For Why Not Higher Score:**

Reviewers objected to many unsubstantiated claims in the paper, which they thought would set a dangerous precedent and/or be misleading for future research. To a lesser extent, the paper leaves some unanswered questions as to where empirical gains come from.

**Justification For Why Not Lower Score:**

N/A

**Metareview: Summary, Strengths And Weaknesses:**

The paper proposes Meta reInforcement Learning via Exploratory clusTering (MILET) as a way to improve knowledge transfer across heterogeneous task distributions in meta reinforcement learning. The method consists in augmenting the VariBAD architecture with a more structured graphical model which teases apart task clusters (discrete) from continuous task embeddings. Similar to MetaCure, separate exploration and exploitation policies are learnt, with novel intrinsic rewards encouraging cluster-level exploration. Various auxiliary losses are also introduced to stabilize training. The method is evaluated across multiple Mujoco environments, and seems to outperform the MetaCure, VariBad, PEARL and RL2 baselines. Qualitative and quantitative measures seem to indicate that the method is able to correctly infer task clusters, and ablations (Table 1 and 2) do a decent job of validating the many design choices of the proposed method.

Whereas most reviewers agree that the paper offers an interesting solution to an important problem, two significant issues prevent us from recommending acceptance at this stage.

While the method does appear effective, reviewers and AC agree that the paper contains many unsubstantiated claims, as first highlighted by [XdEV]. A full list of these can be found here [ https://pastebin.com/3nqByswp ]. At a high-level, empirical performance is fully attributed to the agent’s ability to infer cluster (global) vs task (local) information, or their inability to do so. This is particularly problematic since we know that an explicit representation of cluster and task information is not strictly required and could be captured implicitly by the agent’s belief state (RL2) or as in Bayes Adaptive MDPs, via the posterior distribution over model parameters. While it could be argued that these statements could be easily rectified, we felt that the authors were given ample opportunity to do so during the discussion phase.

The second point of contention is related and compounded by the above. Despite the recent addition of the ablations of Table 2 (Appendix E), reviewers and AC do not understand where the gains are coming from, which points to either a lack of clarity or the need for further ablations and targeted experimentation. For example, why does the ELBO of Eq.4 not collapse the posterior over clusters? To our knowledge there is no extra information available either in the high-level posterior or prior over c. What information could the model infer in c which it cannot express via high-dimensional continuous code z? While the ablations of Table 2 do explore this to some extent, they leave out important controls such as a T-GRU only model, with and without consistency regularization, and with/without intrinsic rewards applied to the inferred task label. In particular, the role of \gamma_h and \gamma_c is not ablated. Does the method work without a negative \gamma_c to encourage exploration over clusters early-on? Could this not be applied directly to $z$ without the need for an explicit representation of clusters? I stress here that such conclusions would in no way diminish the novelty of the paper, but rather help clarify the authors’ contribution.

[Outside the scope of this meta-review, but included in the hopes of improving future revisions] While reviewers appreciated that the paper explored task clusters over both reward functions and dynamics, they also felt that these distributions were limited and overly synthetic in nature. During the discussion, [nQz2] in particular suggested the authors consider richer task distributions where clusters are defined in terms of required skills (e.g. pushing, running) and higher-level semantics (e.g. objects with which to interact).

**Summary Of Ac-Reviewer Meeting:**

All four reviewers were present during the AC-reviewer meeting.

The main points of discussion were as follows:
1. Discussed the main issue of unsubstantiated claims. While this was most actively championed during the discussion phase by [XdEV], all reviewers agreed this was an issue (to varying degree).
2. Whether recent results, especially the ablations of Table 2 sufficiently allayed the reviewers' concern.
3. Discussed various technical points: potential for posterior collapse, susceptibility to cluster misspecification, why method would help when tasks are uniformly distributed (i.e. Ant-U).

The output of 2 and 3 mostly form the basis of the fourth paragraph of the meta-review (see "second point of contention").